# Study of the Protection and Energy Transmission Modes of One Phase Short Circuit to Ground in Inverters

**DOI:** 10.3390/s23198211

**Published:** 2023-09-30

**Authors:** Dezhi Chen, Zhixiang Zhang, Shichong Zhang, Yun Sun, Wenbo Zhao, Wenliang Zhao

**Affiliations:** 1Key Laboratory of Special Motors and High Voltage Electrical Appliances, Ministry of Education, Shenyang University of Technology, Shenyang 110870, China; zhangzhixiang@smail.sut.edu.cn (Z.Z.); zhangshichong@smail.sut.edu.cn (S.Z.); s_vainglory@smail.sut.edu.cn (Y.S.); zhaowenbo@smail.sut.edu.cn (W.Z.); 2School of Electrical Engineering, Shandong University, Jinan 250061, China; wlzhao@sdu.edu.cn

**Keywords:** frequency converter, IGBT and diode, reactor and capacitor, double boost circuit, inverters, short circuit

## Abstract

Research indicates that phase-to-ground short-circuits in a frequency converter can subject the rectifier diode and IGBT to excessive voltage and current, potentially causing damage if the component selection margin during hardware design is insufficient. In order to solve the above problems, this paper studies the design of the LCL filter and ground short circuit problem of the hundred-kilowatt inverter. Firstly, an analytical method for calculating the DC bus capacitance and reactor of the inverter is proposed. The interaction between the DC bus capacitance and the reactor parameters and performance is considered in the implementation process. The parameters of the DC bus capacitor and reactor are given. Secondly, the one-to-ground short circuit of the inverter is studied, and the energy flow mode and mathematical expression of the double boost circuit, considering the influence of the leakage inductance of the power transformer, are given. Based on the above analysis, a method for determining the rectifier diode and IGBT, considering the one-to-ground short circuit of the inverter, is proposed. Finally, a one-hundred-kilowatt inverter is developed, and the corresponding experiments are carried out. The feasibility of the proposed scheme is verified by simulation and experiment.

## 1. Introduction

Due to the vulnerability of power electronic devices and the complexity of their control, the inverter part, especially the inverter part that implements various PWM control strategies, is a weak link in the system that is prone to failure [1]. Therefore, it is of great practical significance to study how to improve the reliability of the inverter and avoid downtime caused by faults.

The existing research on direct current(DC) bus capacitors either focuses on the reliability of DC capacitors from the perspective of harmonic distribution and electro-thermal stress [2] or uses complex control techniques to reduce the root mean square value of DC bus current [3,4]. There is no literature to discuss the design of DC containers; DC reactor can effectively reduce the peak and rising rate of DC fault current. The current research mainly focuses on the design and optimization of medium and high-voltage DC reactors [5], which has little guiding significance for the design of low-voltage DC reactors.

Although domestic and foreign scholars have performed a lot of research on the selection of DC bus capacitance and DC reactor parameters, they have not considered the impact of a single-phase ground short circuit of the inverter. Aiming at the problem of large voltage and large current caused by the single-phase grounding of the inverter, this paper deduces the formula of DC bus capacitance parameters considering voltage temperature and other factors based on the Newton–Raphson method and gives the method of determining the parameters of rectifier diode and IGBT considering single-phase short circuit. A 380 V/110 kW inverter is fabricated, and the relative ground short circuit test is carried out. The simulation and experimental results show the accuracy and feasibility of the proposed method.

## 2. Design of Capacitor and DC Reactor in DC Bus

All units of measurement involved in this paper and their abbreviations are shown in Table 1.

Table 2 shows the system parameters of the 380 V/110 kW frequency converter to be designed in this paper.

Figure 1 shows the topology of the AC-DC-AC voltage-type frequency control device. It mainly includes a rectifier diode, LC filter, soft start circuit, inverter, load, etc. As shown in Figure 1, for the small and medium power frequency control devices, the system only needs to install the incoming reactor. However, when the system power is large, the DC reactor will be connected in series on DC should be limited to a specific value to maintain the continuity of the current and reduce the current ripple. Enhancing the mitigation of input current waveform distortion arising from capacitor filtering, improving the power factor, and reducing and preventing damage to the rectifier bridge and capacitor overheating due to excessive pulse current [6,7,8,9].

### 2.1. DC Bus Capacitance Design

#### 2.1.1. DC Bus Capacitance Calculation

The DC bus capacitance is adopted in the inverter to absorb the fluctuation of the wave so that the output of the system is more steady [10,11]. This paper presents a comprehensive design for the DC Bus Capacitor from the aspects of voltage, volume, temperature, power consumption, and lifetime. Figure 2 illustrates the design process of a DC bus capacitor using the Newton–Raphson approach.

First, the charging and discharging process of the capacitor at different times is analyzed according to the system structure and parameter values. Secondly, the capacitance value is calculated using Newton–Raphson’s method, and then the ripple current is calculated. Finally, the stability of the system is judged.

When the DC bus capacitance is large enough, the waveform of the diode rectified output voltage in one cycle is shown in Figure 3. Among them, *U_rec_* and *U_dc_* are the output voltage waveforms of three-phase uncontrolled rectifier circuits with or without DC bus capacitance.

According to the output characteristics of a three-phase uncontrolled rectifier circuit in one rectifier cycle:(1)$$\left\{\begin{array}{l} U_{rec}=\begin{array}{cc} \max\left(\sqrt{2}U_{N}\cos\omega t,\sqrt{2}U_{N}\cos(\omega t-\frac{\pi }{3})\right) & \end{array}\\ I_{dc}=\frac{P_{o\_dc}}{U_{c}}+C_{dc}\frac{dU_{c}}{dt} \end{array}\right.$$
where *I_dc_*, *U_c_*, *P_o_dc,_* and *C_dc_* are DC bus current, bus voltage, bus capacitance output power, and bus capacitance, respectively.

Stage (*t_0_*~*t_1_*):

The power required by the load is provided by the three-phase input and the bus capacitor. At this time, the capacitor is in a discharge state.

When *t* = *t*_1_, *I*_c_ = −*I*_load_, that is:(2)$$-\sqrt{2}C_{dc}U_{N}\omega \sin\omega t=-\frac{P_{o\_dc}}{\sqrt{2}U_{N}\cos\omega t}$$
(3)$$t_{1}=\frac{\arcsin\frac{P_{o\_dc}}{\omega C_{dcC}U_{N}^{2}}}{2\omega }$$

Stage (*t*_1_~*t*_2_):

The power required by the load is completely provided by the bus capacitor, and the bus capacitor is in a discharged state.
(4)$$C_{dc}\frac{dU_{c}}{dt}=-\frac{P_{o\_dc}}{U_{c}}$$

To solve the differential Equation:(5)$$U_{c}=\sqrt{{\left(\sqrt{2}U_{N}\cos\omega t_{1}\right)}^{2}-\frac{2P_{o\_dc}(t-t_{1})}{C_{dc}}}$$

When *t* = *t*_1_, that is:(6)$$U_{c}=\sqrt{2}U_{N}\cos(\omega t-\frac{\pi }{3})$$

At this time, the bus voltage drop percentage and *t*_2_ can be obtained, as shown in Equations (7) and (8).
(7)$$\alpha =1-\frac{U_{c}(t_{2})}{\sqrt{2}U_{N}}$$
(8)$$t_{2}=\frac{\frac{\pi }{3}-\arccos(1-\alpha )}{\omega }$$

Stage (*t*_2_~*t*_3_):

The three-phase input supplies power to the load and the bus capacitors. At this time, the capacitor begins to charge.
(9)$$\left\{\begin{array}{l} U_{c}=\sqrt{2}U_{N}\cos(\omega t-\frac{\pi }{3})\\ I_{c}=-\sqrt{2}C_{dc}U_{N}\omega \sin(\omega t-\frac{\pi }{3}) \end{array}\right.$$

When the capacitance of the bus is not enough to fully absorb the ripple current on the bus:(10)$$t_{1}\geq \frac{\pi }{6\omega }$$

Substituting Equation (3) into Equation (10), get:(11)$$\frac{P_{o\_dc}}{\omega C_{dc}U_{N}^{2}}\geq \sin\frac{\pi }{3}$$

When the circuit parameters meet the conditions of Equation (11), *U_c_* = *U_rec_* is valid for the entire circuit working process. At this time, the percentage of bus voltage drop is:(12)$$\alpha =1-\cos\frac{\pi }{6}$$

Now suppose that the load current is completely obtained from the bus capacitance at time *t*_0_, when *t*_1_ = *t*_0_, according to Equation (5):(13)$$U_{c}=\sqrt{{\left(\sqrt{2}U_{N}\right)}^{2}-\frac{2P_{o\_dc}t}{C_{dc}}}$$

When *t* = *t*_2_, combining Equations (7) and (13) can have:(14)$$C_{dc}=\frac{P_{o\_dc}\left[\frac{\pi }{3}-\arccos(1-\alpha )\right]}{\omega U_{N}^{2}(2\alpha -\alpha^{2})}$$

In this situation, the discharge time is longer than the actual discharge time. Therefore, the calculated value of *C_dc_* obtained by Equation (14) is greater than the actual value. If capacitance calculated by Equation (14) is the max of the bus capacitance *C*_dc_max_, then the initial search interval of Newton–Raphson method can be determined as [0, *C*_dc_max_]. It is assumed that the bus capacitance can completely absorb the ripple current and then leave a little margin, taking:(15)$$\alpha =(1-\cos\frac{\pi }{6})\times 98.5\%=13.2\%$$

According to the above principle, the bus capacitance is determined as 5500 μF, which is composed of six 3900 μF/450 V capacitors in series and parallel.

#### 2.1.2. Calculation of Bus Capacitance Loss

The loss of capacitance consists of two parts: one is caused by equivalent series resistance(ESR), and the other part is caused by leakage current [12]. Since the leakage current loss is usually much smaller than the ESR loss, it can be ignored in the calculation.

The DC bus capacitor ripple current is [13]:(16)$$I_{r_{-}Cdc}=\sqrt{{\left\{I_{dc}\sqrt{2M\left[\frac{\sqrt{3}}{4\pi }+\cos^{2}\varphi \left(\frac{\sqrt{3}}{\pi }-\frac{9}{16}M\right)\right]}\right\}}^{2}+{\left[\alpha \cdot C_{DC}\cdot \sqrt{2}U_{N}\sqrt{f_{c\_ms}\left(\frac{1}{t_{c}}+\frac{1}{t_{f}}\right)}\right]}^{2}}$$
where *M* is the PWM modulation ratio, cos*φ* is the power factor, *f*_c_rms_ is the pulse frequency, *t_c_* is the charging time of the DC bus capacitor, and *t_f_* is the discharge time of the DC bus capacitor.

The loss of bus capacitance is:(17)$$P_{Cdc}=EI_{r\_Cdc}^{2}ESR_{Cdc}={\left(\frac{I_{r\_Cdc}}{K_{FC}}\right)}^{2}ESR_{Cdc}$$
where *EI_r_Cdc_* is the equivalent ripple current flowing through the bus capacitor, *ESR_Cdc_* is the ESR of the bus capacitor, *I_r_Cdc_* is the ripple current flowing through the bus capacitor, and *K*_FC_ is the frequency coefficient of the bus capacitor.

After three-phase rectification and filtering, the fundamental frequency of the current is six times the input frequency. For this system, the bus capacitance frequency coefficient *K*_FC_ = 1.1 is calculated, and the loss of the bus capacitor is calculated as shown in Table 3.

#### 2.1.3. Life Evaluation of Bus Capacitance

According to the technical documents of the Hitachi capacitor, considering the ripple current and ambient temperature, the capacitor life is estimated as:(18)$$L=L_{r}\times 2^{\frac{T_{0}-T}{10}}\times K^{\frac{\Delta T_{0}-\Delta T}{10}}$$
where *L_r_* is the service life of the bus capacitor at the rated ripple current and the highest operating temperature, *T*_0_ is the highest guaranteed temperature, *T* is the actual working temperature, Δ*T*_0_ is the allowable temperature rise of the capacitor center at the highest operating temperature, Δ*T* is the actual center temperature rise, *K* is acceleration factor of ripple current (*K* = 2 is the allowable range of ripple current, *K* = 4 is beyond the ripple current range).
(19)$$\Delta T={\left(\frac{EI_{r}}{I_{r\_C}}\right)}^{2}\Delta T_{0}={\left(\frac{I_{r}}{K_{FC}I_{r\_C}}\right)}^{2}\Delta T_{0}$$
where *EL_r_* is the equivalent ripple current flowing through the capacitor, *I_r_C_* is rated ripple current at max working temperature, and *I_r_* is the ripple current flowing through the capacitor.

According to the ambient temperature of the system, it is assumed that the operating temperature of the capacitor under rated load is 50 °C, and it is 60 °C under heavy load. The calculation results of the bus capacitor life are shown in Table 3.

According to the data analysis in Table 3, it can be seen that under load, the loss of single bus capacitance and total bus capacitance is the highest, and under overload, the service life of the capacitor bus is the shortest.

### 2.2. DC Reactor Design

#### 2.2.1. Color/Grayscale Figures

The DC reactor is connected before the bus capacitor, as shown in Figure 1. The main function of the DC reactor is to improve the input current distortion caused by the capacitor filtering, improve the power factor, and reduce and prevent the rectifier bridge damage and capacitor overheating caused by the inrush current.

The inductance value of the DC reactor should not be too large or too small. An excessively large induction value will cause a large voltage drop on the reactor, resulting in a low DC bus voltage. In a variable frequency speed control system with a constant power load, instability will occur. The too-small inductance will cause excessive ripple current to flow through the bus capacitor, causing greater loss, affecting the service life of the capacitor, and also causing too-high Total Harmonic Distortion (THD) and reducing the power factor of the system.

In this paper, the upper limit of the inductance value is determined by ensuring the system stability and the voltage on the reactor is not large, and the lower limit of the inductive value is determined by ensuring the life of the bus capacitor. By using the small signal analysis method, the characteristic Equation of the circuit transfer function can be deduced as follows:(20)$$s^{2}+\left(\frac{R_{dc}}{L_{dc}}-\frac{P_{o\_dc}}{C_{dc}{\bar{V_{dc}}}^{2}}\right)s+\frac{1}{L_{dc}C_{dc}}\left(1-\frac{R_{dc}P_{o\_dc}}{{\bar{V_{dc}}}^{2}}\right)=0$$
where *R_dc_* is the DC equivalent resistance on the rectifier filter circuit, *L_dc_* is DC inductance, and is the average value of DC bus voltage.
(21)$$\bar{V_{rec}}=\bar{V_{dc}}+\frac{P_{o\_dc}}{\bar{V_{dc}}}R_{dc}$$

$\bar{V_{rec}}$ is the average value of voltage after rectification. The results are as follows:(22)$$\bar{V_{dc}}=\frac{\bar{V_{rec}}+\sqrt{{\bar{V_{rec}}}^{2}-4P_{o\_dc}R_{dc}}}{2}$$

In general:(23)$$\left|\frac{R_{dc}P_{o\_dc}}{{\bar{V_{dc}}}^{2}}\right|<<1$$

According to Equation (20), to ensure the system’s stability, it shall meet the following requirements:(24)$$\frac{R_{dc}}{L_{dc}}-\frac{P_{o\_dc}}{C_{dc}{\bar{V_{dc}}}^{2}}>0$$

According to experience, the value of a DC reactor generally does not exceed three times the 3% impedance inductance of the same inverter input AC reactor. Considering the overload, taking the DC equivalent resistance RDC = 0.04 Ω, and calculating Equation (25), the upper limit of the inductance value of the DC reactor can be obtained, as shown in Table 4.
(25)$$L_{dc}\leq \frac{3\times 3\%V_{N}}{\sqrt{3}\times 2\pi f_{N}I_{N}}$$

The longevity of an electrolytic capacitor predominantly relies upon the rated lifespan of the capacitor itself, the ambient temperature to which it is exposed, and the magnitude of ripple current traversing through it. Additionally, the inductor within the circuit significantly impacts the amplitude of the ripple current flowing through the bus capacitor. Consequently, ensuring a specific inductance value becomes imperative to attain the desired operational lifespan of the bus capacitor, thereby averting premature failures.
(26)$$I_{r}=K_{FC}I_{r\_C}\sqrt{1-\frac{10}{\Delta T_{0}}\log_{K}\left(\frac{L}{L_{r}\times 2^{\frac{T_{0}-T}{10}}}\right)}$$

When the design life *L* is known, the maximum ripple current that the capacitor can withstand can reach Equation (26), and the minimum inductance of the DC reactor can be determined according to the maximum ripple current.

The circuit model, shown in Figure 1, corresponds to a nonlinear second-order differential equation, which is difficult to solve, and it is impossible to determine the initial values of the variables in the steady state of the circuit. This paper uses the constant step Euler algorithm to simulate the circuit. Finally, average voltage, peak-to-peak ripple voltage, root mean square (RMS) voltage, ripple current of the bus capacitor, and RMS current, average current, and ripple current of the DC reactance can be acquired.

In the simulation calculation process, since the sampling period is much shorter than the operating period of the circuit, the rectified voltage can be approximated as a constant voltage source within a sampling period, and the load is approximated as a resistance, which can not only greatly simplifies the mathematical model of the circuit, but also does not have much impact on the calculation results.

Thus, the circuit model can be expressed as:(27)$$\left\{\begin{array}{l} C_{dc}\frac{dV_{C}}{dt}+\frac{V_{C}}{R_{load}}=I_{L}\\ L_{dc}\frac{dI_{L}}{dt}+I_{L}R_{dc}+V_{C}=V_{rec} \end{array}\right.$$
where *I_L_* is the current flowing through the DC reactor, $R_{load}=\frac{V_{C}^{2}(t)}{P_{O\_DC}}$ is load equivalent resistance, and *V_rec_* is the equivalent voltage after rectification.

To solve the system of differential equations, acquire:

When *L_DC_* is known, the simulation calculation can be carried out by Equation (28) and Equation (29), and the *V_C_* and *I_L_* of each transient can be acquired, and then calculating the required voltage and current data. If using dichotomy to find *L*_DC_ in the range of [0, *L*_DC_Max_], which is in accordance with the calculation result of Equation (29), the lower limit *L*_DC_Min_ of the inductance value of the DC reactor is received. Considering the rated load, the lower limit of the inductance value of the DC reactor can be calculated according to the above method, as shown in Table 5.
(28)$$\left\{\begin{array}{l} V_{C}(t)=R_{load}x_{7}V_{rec}+\frac{\left[R_{load}x_{7}(x_{3}-x_{6})V_{rec}+(x_{4}+x_{6})V_{C}(0)-2R_{load}x_{1}I_{L}(0)\right]}{2x_{6}}e^{-\frac{x_{3}+x_{6}}{x_{5}}t}-\frac{\left[R_{load}x_{7}(x_{3}+x_{6})V_{rec}+(x_{4}-x_{6})V_{C}(0)-2R_{load}x_{1}I_{L}(0)\right]}{2x_{6}}e^{-\frac{x_{3}-x_{6}}{x_{5}}t}\\ I_{L}(t)=x_{7}V_{rec}+\frac{\left[x_{7}(x_{4}-x_{6}-2x_{2}\frac{R_{load}}{R_{dc}})V_{rec}+2x_{2}\frac{V_{C}(0)}{R_{dc}}-(x_{4}-x_{6})I_{L}(0)\right]}{2C_{6}}e^{-\frac{x_{3}+x_{6}}{x_{5}}t}-\frac{\left[x_{7}(x_{4}+x_{6}-2x_{2}\frac{R_{load}}{R_{dc}})V_{rec}+2x_{2}\frac{V_{C}(0)}{R_{dc}}-(x_{4}+x_{6})I_{L}(0)\right]}{2x_{6}}e^{-\frac{x_{3}-x_{6}}{x_{5}}t} \end{array}\right.$$
where *C*_1_~*C*_7_ are all constants, and:(29)$$\left\{\begin{array}{l} x_{1}=\frac{L_{dc}}{R_{dc}}\\ x_{2}=R_{load}C_{dc}\\ x_{3}=x_{1}+x_{2}\\ x_{4}=x_{1}-x_{2}\\ x_{5}=2x_{1}x_{2}\\ x_{6}=\sqrt{x_{4}^{2}-2x_{5}\frac{R_{load}}{R_{dc}}}\\ x_{7}=\frac{1}{R_{dc}+R_{load}} \end{array}\right.$$

It can be seen from Table 5 that in the case of 200% overload in heavy load application, the lower limit value of *L*_DC_ still cannot meet the life requirement of bus capacitor, but the time of 200% overload in heavy load application is very short and can be ignored. According to the real situation, the inductance value of a DC reactor is 230 μH.

#### 2.2.2. Electromagnetic Design of DC Reactor

According to the actual situation, the silicon steel sheet 35Z155 of Nippon Steel Co., Ltd., Tokyo, Japan. is selected as the core material of the reactor, and EI type core of EI133.2 is selected, as shown in Figure 4, and core parameters are shown in Table 6.

According to the requirements of reactor inductance and current, 220 silicon steel sheets are used to form the EI core. The thickness of the core is:(30)$$T=Thick_{Steelsheet}N_{Steelsheet}$$

According to Equation (30), the core thickness of the DC reactor is *T* = 77 mm, the core size of the DC reactor is 133.2 mm × 111 mm × 77 mm, and the relevant parameters of the core can be calculated as follows: *A*_e_ = 3.4188 × 103 mm^2^; *L*_e_ = 266.4 mm; *V*_e_ = 9.1077 × 105 mm^3^; *A*_CW_ = 1.4785 × 103 mm^2^; *W*_t_ = 6.9674 kg; *P*_CL_ = 10.1724 W (50 Hz, 1.7 T).

According to the law of ampere and flux:(31)$$N_{turns}=ceil\left(\frac{LI_{peak}}{B_{m}A_{e}}\right)$$
where *I_peak_* is the peak current under rated load, and *B_m_* is the max working flux density.

Take *B*_m_ = 1.4 T, and calculate Equation (31) to obtain the number of turns of DC reactor winding *N_turns_* = 8. The calculated length of the air gap is:(32)$$l_{g}=\mu_{0}A_{g}\left(\frac{N_{turns}^{2}}{L}-\frac{L_{e}}{\mu_{e}A_{e}}\right)$$
where *A_g_* is the cross-sectional area of the magnetic circuit considering the air gap, and *μ_e_* is the permeability of the core material, which can usually be calculated from the *B* = *f*(*H*) curve of the core material.

When the air gap is large, the edge magnetic flux cannot be ignored. The cross-sectional area of the magnetic circuit considering the edge magnetic flux is:(33)$$A_{g}=A_{e}+\left[\frac{4m(C+T)}{\pi (l_{g}+m)}+0.52(C+T)+0.308l_{g}+m\right]l_{g}$$
where *l_g_* is the length of the air gap, and *m* is the width of edge flux diffusion.

Perform iterative operation on Equations (32) and 33), making *A_g_* = *A_e_*, calculating *l_g_* using Equation (32), and then substitute the calculated *l_g_* into Equation (33) to obtain a new *A_g_*, then substitute the new *A_g_* into Equation (32), to cycle until *l_g_* no longer has a big change, then calculate the air gap length that the reactor needs. If *m* = *l_g_* and calculated according to the above algorithm, the air gap length of DC reactor *l_g_* = 2.6 mm can be acquired.

Core saturation verification using Equation (34):(34)$$B_{\max}=\frac{LI_{\max}}{N_{turns}A_{e}}$$

When *I*_max_ = 386 A, the maximum operating flux density of the core *B*_max_ = 1.6188 T. When the current reaches the maximum value, the inductance will drop, but it can meet the design requirements.

According to the set current density, the current-carrying area required by the copper wire can be calculated as follows:(35)$$S_{turns}=\frac{I_{rms}}{J}$$
where *S_turns_* is the current-carrying area required for single-turn copper wire, and *J* is the current density.

Take *J* = 3.57 A/mm^2^, calculate Equation (35), and obtain the current-carrying area required for turning copper wire *S_turns_* = 71.4297 mm^2^.

Therefore, according to the actual situation, 60 mm × 0.3 mm copper foil is selected for winding the reactor. When using four layers of copper foil for a single turn, the current density can meet the requirements. The current density is 3.47227 A/mm^2^.

The formula for calculating the window occupancy rate is
(36)$$K_{CW}=\frac{S_{turns}N_{turns}}{A_{CW}}$$

Calculate Equation (36), *K_CW_* = 0.3896.

In summary, considering the thickness of the core, insulation layer, and wire package, it can be estimated that the size of the DC reactor is approximately 133.2 mm × 111 mm × 101.6 mm.

The loss of the DC reactor is composed of copper loss and iron loss. Copper loss refers to the loss of current through the copper wire, and iron loss refers to the loss in the iron core. This relationship can be expressed as:(37)$$P_{tot}=P_{Cu}+P_{Fe}$$
where *P_Cu_* is the copper consumption of the DC reactor, and *P_Fe_* is the iron consumption of the DC reactor.

The series equivalent resistance of the DC reactor is:(38)$$ESR_{L}=\rho \frac{MLT\times N_{turns}}{S_{turns}}$$
where is the resistivity of copper wire and *MLT* is the average length of winding.

The copper consumption of the DC reactor is:(39)$$P_{Cu}=I_{RMS}^{2}ESR_{L}$$

The iron consumption of the DC reactor is:(40)$$P_{Fe}=P_{CL}{\left(\frac{f}{50}\right)}^{K_{f}}{\left(\frac{B_{AC}}{1.7}\right)}^{K_{B}}$$
where *f* is the frequency of the ripple current flowing through the DC reactor, *B_AC_* is the AC magnetic flux density, *K_f_* is the frequency coefficient, and *K_B_* is the magnetic flux density coefficient.

Among them, the AC flux density is:(41)$$B_{AC}=\frac{LI_{P-P\_Ldc}}{2N_{turns}A_{e}}$$
where *K_f_* and *K_B_* can be calculated based on the data provided in the core material specifications. For 35Z155, the frequency coefficient *K_f_* = 1.5067, and the magnetic flux density coefficient *K_B_* = 2.8969.

For a three-phase uncontrolled rectifier circuit, ripple current frequency *f* = 300 Hz.

Equations (37)–(41) can be used to acquire the equivalent series resistance and loss of each part of the DC reactor, as shown in Table 7.

The magnetic flux density distribution of the DC reactor under rated load current is shown in Figure 5. The magnetic flux density distribution of the DC reactor under overload conditions is shown in Figure 6.

It can be seen from Figure 5 and Figure 6 that the flux density of the DC reactor is 1.1 T under rated load and 1.3 T under overload, which is not saturated and meets the system requirements.

## 3. Simulation Analysis of Bus Capacitance and DC Reactor under Normal Conditions

Figure 7 shows the simulation block diagram of the inverter LC system based on MATLAB/Simulink software. It mainly includes an equivalent power transformer, rectifier circuit, LC filter circuit, load, etc. In this paper, the simulation switching frequency is 5 kHz. Figure 8 shows the bus voltage waveform. Figure 9 shows the ripple current waveform of bus capacitance. Figure 10 shows the output current waveform of the frequency converter.

According to Figure 8, it can be seen that the selection of system bus capacitance and DC reactor is reasonable, the output current waveform is stable, and the correctness of the proposed scheme is verified.

Further analysis: without adding a reactor, considering the influence of a power transformer, when a phase-to-ground short circuit occurs (for instance, phase K is closed, phase W to ground short circuit), it is possible to explore the machine. The energy transmission mode when a phase-to-ground short circuit occurs in the frequency control device is analyzed.

### 3.1. DC Bus Capacitance Design

In instances where a ground short circuit (K_1_ closed) arises, and the output current assumes a positive direction, an isolated circuit is configured, comprising the maximum phase voltage *V*_r_, along with its corresponding diodes D1, VT1, VD2, and capacitor C.

Figure 9 shows a diagram of the VT1 rotor with energy transfer at shutdown.

Upon the activation of VT1, the *L*_kr_ voltage is charged from the power supply, subsequently leading to a rise in output current. Conversely, upon the deactivation of VT1, an instantaneous change occurs in the *L*_kr_ current, wherein the unobstructed circuit path encompasses D1, C, and VD2. During this phase, C charges, resulting in an augmentation of the bus voltage.

### 3.2. VT2 Turns On and Off Energy Transmission Modes

Figure 10 depicts energy transfer modes when VT2 is alternately switched on and off.

Upon the current of *L*_kr_ attaining a decrease to 0, a boost circuit is formed, comprising the lowest phase voltage *V*_r_, the leakage inductance *L*_kt_, along with the corresponding diode D6, DC reactor L, VT2, VD1, and capacitor C. When VT2 is switched on, *V*_t_, *L*_kt_, D6, *L*, and VT2 form a circuit, *L*_kt_ is charged from the supply, and the output current is reversed. When VT2 is switched off, the current L, *L*_kt_ is not immediately switched, the current is frozen by D6, *C,* and VD1 is charged, and the voltage of the bus is increased.

## 4. Simulation of Single-Phase Earthing Short Circuit Scenarios for Inverters

The incorporation of a DC reactor preceding the bus capacitor serves as a primary measure to ameliorate the adverse effects induced by abnormal fluctuations in the capacitor filter input current. This arrangement aims to enhance power factor, mitigate the potential hazards of rectifier bridge damage, and prevent overheating of the capacitor caused by inrush currents. There are generally two structures: one is to connect a DC reactor in series with the cathode of the DC bus, and the other is to connect a DC reactor in series with the anode.

### 4.1. A DC Reactor on the Cathode of the DC Bus

Figure 11 illustrates the PWM waveform, output current, DC bus voltage wave, and capacitance current wave of the IGBT. Additionally, Figure 12 presents the output voltage and output current characteristics of the bridge arm on the rectifier. Furthermore, Figure 13 depicts the output voltage and current behaviors of the IGBT bridge arm.

Upon the occurrence of an earthing short circuit in the inverter system, there is an evident rise in the currents *I*_A_ and *I*_CAP_, along with a corresponding increase in the DC bus voltage, denoted as Vdc. The voltage *V*_dc_ exhibits a transient behavior characterized by abrupt ascents and descents, a phenomenon attributed to the ESR of the capacitor.

Because of the way it transmits the earth’s short circuit, the rectifier will not work properly under the condition of earthing. When some diodes turn on, the Corresponding diode turns off. When D6 and VT2 are turned on, *V*_t_ < *V*_r_ and *V*_t_ < *V*_s_, the capacitor is the same as the capacitor and ground, so *V*_D3_ = −*V*_t_ + *V*_dc_. Actually, due to parasitic effects, the situation becomes much more serious.

The output voltage and the current in the AC–DC converter are high, so IGBT is easily destroyed due to overheating.

### 4.2. A DC Reactor on the Anode of the DC Bus

Figure 14 presents graphical representations of the PWM waveform, the output current, the DC bus voltage wave, and the capacitance current wave associated with the IGBT. Subsequently, Figure 15 provides an illustrative depiction of the output voltage and output current characteristics exhibited by the bridge arm within the rectifier. Furthermore, Figure 16 offers a detailed account of the output voltage and current attributes of the IGBT bridge arm.

As illustrated in Figure 14, Figure 15 and Figure 16, a substantial current flow is observed using the bridge rectifier arm, exhibiting a notable disparity when compared to the negative state of the DC bus in conjunction with the DC reactor.

### 4.3. A DC Reactor on the Anode and Cathode of the DC Bus

Figure 17 depicts the PWM waveform of the IGBT, the output current, the DC bus, and the capacitor current. Similarly, Figure 18 showcases the output voltage and output current characteristics of an abridge arm on a rectifier. Furthermore, Figure 19 presents the output voltage and current behaviors specific to the IGBT bridge arm.

If, by the depictions in Figure 11, Figure 14 and Figure 17, the original reactor’s value is halved and symmetrically positioned on the positive and negative DC buses, a more equilibrated distribution of current through the devices is achieved, thereby contributing to an enhanced operational state.

Simulation results for each type of inverter are given in Table 8. If the DC inductor is divided into two reactors with the same value, one on the positive bus and the other on the negative bus, it can have a better common mode filtering effect and improve EMC performance. When the output of the system is short-circuited to the ground, since there are reactors on the positive and negative bus bars, it can evenly protect the up and downpipes of the rectifier bridge and enhance the reliability of the system. Thus, the DC inductor of this system is designed as two reactors of 115 A/250 μH, which are, respectively, placed in the positive and negative buses.

## 5. Design of Rectifying Diode

When the output of the inverter is short-circuited to the ground, and the bus voltage rises, the reverse voltage drop that the rectifier may bear is:(42)VR_Rec=VDC_max+23KOVVN

Therefore, the reverse repetitive peak voltage *V*_RRM_ of the rectifier should be higher than the calculation result of Equation (42), and there shall be a margin to obtain:(43)VRRM_Rec≥VDC_max+23KOVVN

Considering the maximum bus voltage of the frequency converter and the maximum input grid voltage, according to Equation (43), the minimum repetitive peak reverse voltage *V_RRM_Rec_* of the rectifier is 1349.1 V. The average current flowing through each rectifier is:(44)IAV_Rec=IAV_Ldc3

The RMS current flowing through each rectifier is:(45)IRMS_Rec=IRMS_Ldc3

Additionally, the rectifier diode should meet the following requirements:(46)IAV_D≥1.5IAV_Rec_ovd
(47)IRMS_D≥1.5IRMS_Rec_ovd

Table 9 is the effective average current and rated effective current of the rectifier diode calculated by the formula Equations (44)–(47).

Based on the above, this system uses the thyristor diode module of Infineon model TD180N16KOF as the rectifier of the machine, which is composed of three modules.

Thus, in future design, the impact of short-circuit to ground should be considered. The rectifier module should be a module with a withstand voltage of more than 1600 V (the voltage level of the currently selected module is 1600 V), and the diode in series with the soft starting resistor should also select the diode with a voltage of more than 1600 V (the voltage level of the currently selected diode is 1000 V). If possible, the DC reactor should be designed as a double wound Group, one for each positive and negative bus.

## 6. Inverter Single-Phase Ground Short Circuit Experiment

In this paper, the single-phase-to-ground short-circuit experiment of the inverter is carried out. The main experimental devices include three parts: a permanent magnet synchronous motor, inverter, and oscilloscope. Figure 20 shows the experimental circuit diagram. The inverter adopts an AC-DC-AC structure, and Figure 21 is the experimental site.

The oscilloscope, current, and voltage measurement device used in this experiment are shown in Figure 22.

Table 10 lists the measurement conditions of this experiment.

According to the experimental circuit shown in Figure 20, the experimental system in Figure 21 is built. As shown in Figure 21, a contactor is connected between the output end of one phase of the inverter and the earth. By adding or removing 220 V voltage to the control end of the contactor, the on-off of the contactor can be controlled. The specific experimental steps are as follows:

Step 1: Set the oscilloscope channel 1 as the upper bridge arm drive signal with the ground short circuit bridge arm, channel 2 as the short circuit phase output current, channel 3 as the DC bus voltage, and channel 4 to the diode reverse tube voltage drop.

Step 2 Set the oscilloscope channel 3 as the trigger signal.

Step 3: Let the contactor in the disconnected state, and let the inverter output a certain frequency of alternating current and stable.

Step 4: Suddenly close the contactor so that a phase of the inverter outputs a short circuit with the ground.

Step 5: Collect data.

Table 11 shows the experimental results of the one-to-ground short circuit of the inverter.

As shown in Table 2 and Table 11, the simulation and experiment results show that there is a sudden rise and fall of Vdc caused by the capacitance of the ESR. When D6 and VT2 turn on, V_T_ is the lowest in the three phases, and the bus capacitor anode potential is equal to the earth, so D3 has a high voltage of −V_T_ +V_DC_ when L is charged by −V_T_.

The experimental results are shown in Figure 23. Things will be worse practically due to the parasitic parameters. It can be seen from the experiment results that when the output current increases to 433.3 A, the bus voltage increases to 829 V, and the diode reverse voltage rises to 1604 V. The current flowing through all devices will be balanced and reduced if the DC reactor with half the value of the original reactor is placed on the positive and negative sides of the DC bus, respectively.

On the positive and negative of the bus installed a DC reactor, when inverter earthing short circuit happens, V_DC_ and I_A_, etc., decrease significantly. It has proved to be an effective way to avoid damage due to short circuits caused by the device.

## 7. Conclusions

Aiming at the fault caused by a phase-to-ground short circuit of the frequency converter, this paper systematically studies the LC filter, rectifier diode, and a phase-to-ground short circuit of the 380 V/110 kW frequency converter.

(1)Based on the Newton–Raphson method, the DC bus capacitance of the frequency converter is analyzed and designed. By building a power electronic model of the power supply state of the frequency converter, the expression of DC bus capacitance at different times and the calculation results of loss and life are given.(2)Under the two conditions of ensuring the stability of the system and ensuring the voltage on the reactor not to be too large, the power electronic modeling of the DC reactor is carried out, and the calculation expression of inductance value, magnetic field distribution, and loss calculation process of DC reactor are given.(3)This paper analyzes the one-phase earthing short of inverters considering the effect of leakage inductance of the power transformer, establishes a double boost circuit and introduces its energy transmission process, and comes to a conclusion that the main reasons the DC bus voltage rises are the increase in diode reverse voltage drop caused by the charge and discharge of DC reactor as well as ESR. The current and voltage through rectifiers, DC bus, and diodes decrease a lot by installing DC reactors on both sides of the DC bus when earthing short happens. Simulation and experimental results verify the effectiveness of the proposed scheme.

## Figures and Tables

**Figure 1 sensors-23-08211-f001:**
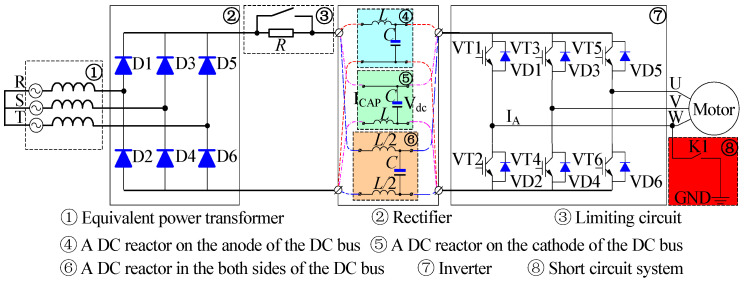
The topology of the inverter.

**Figure 2 sensors-23-08211-f002:**
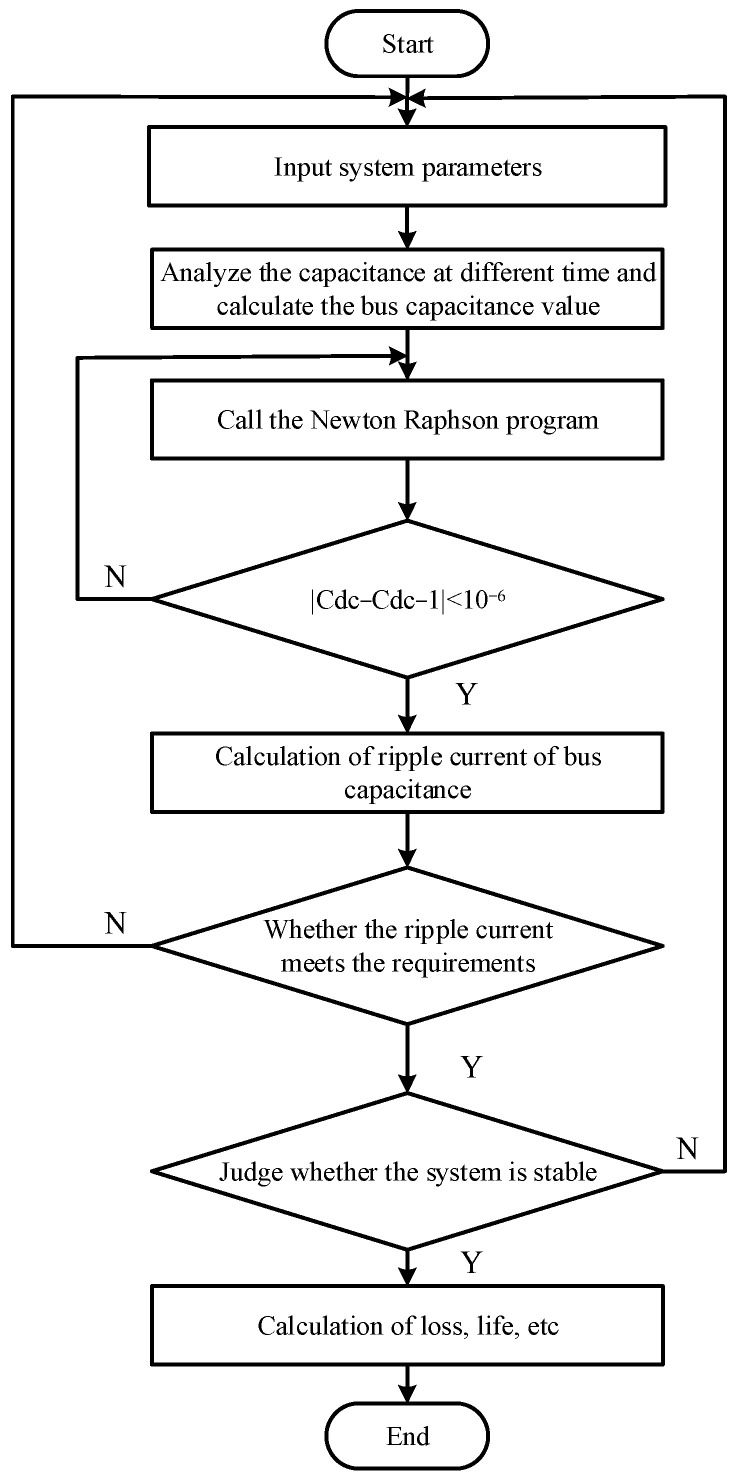
The calculation process of bus capacitance parameter.

**Figure 3 sensors-23-08211-f003:**
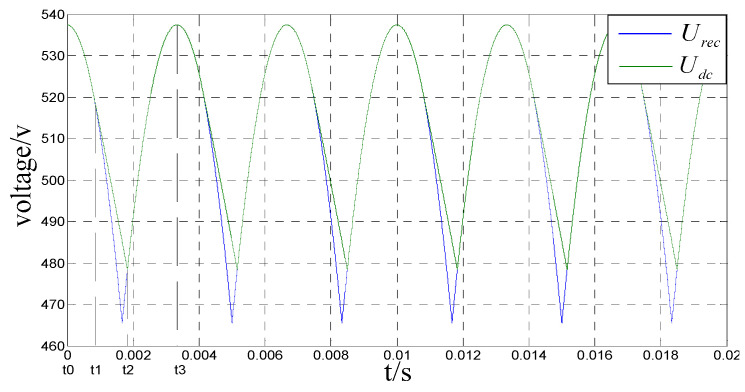
Waveform in rectified output cycle.

**Figure 4 sensors-23-08211-f004:**
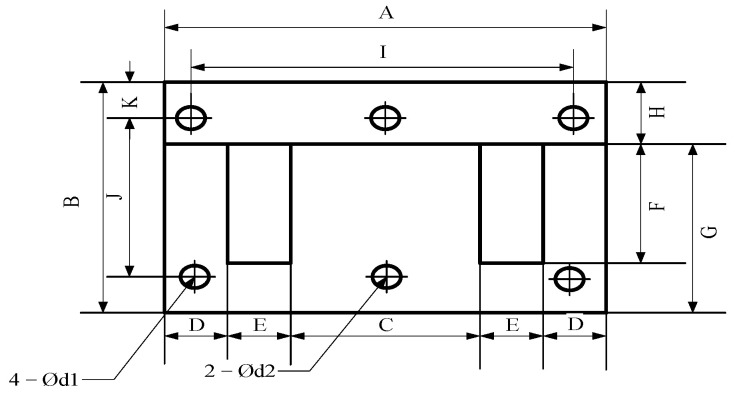
EI-type core structure.

**Figure 5 sensors-23-08211-f005:**
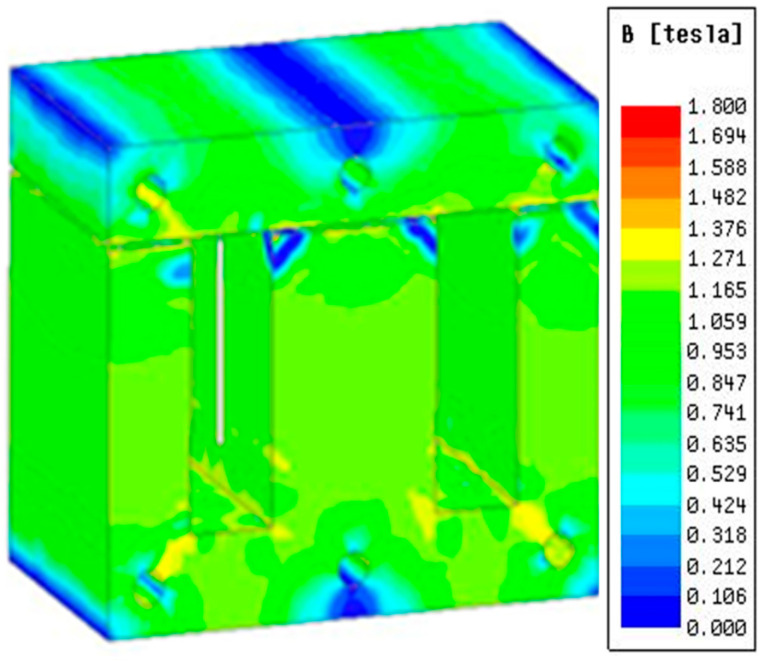
Flux density under rated load.

**Figure 6 sensors-23-08211-f006:**
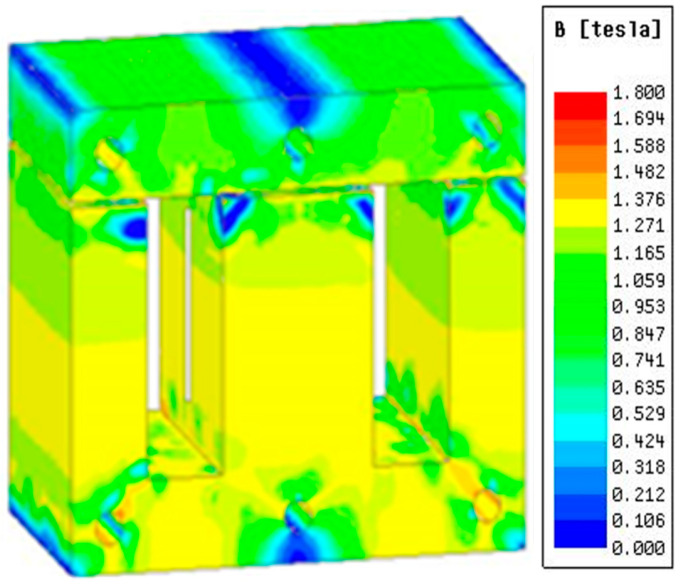
Flux density under overload.

**Figure 7 sensors-23-08211-f007:**
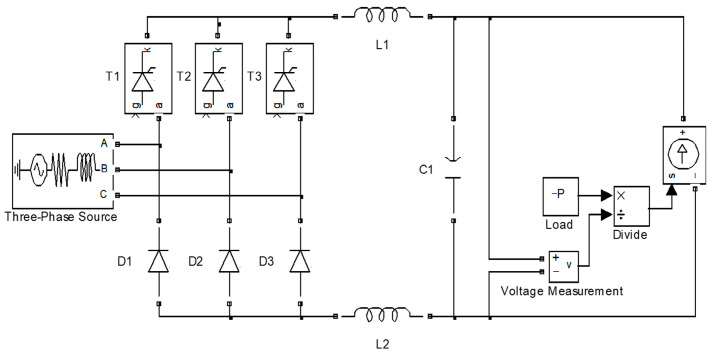
Simulation block diagram of inverter LC system.

**Figure 8 sensors-23-08211-f008:**
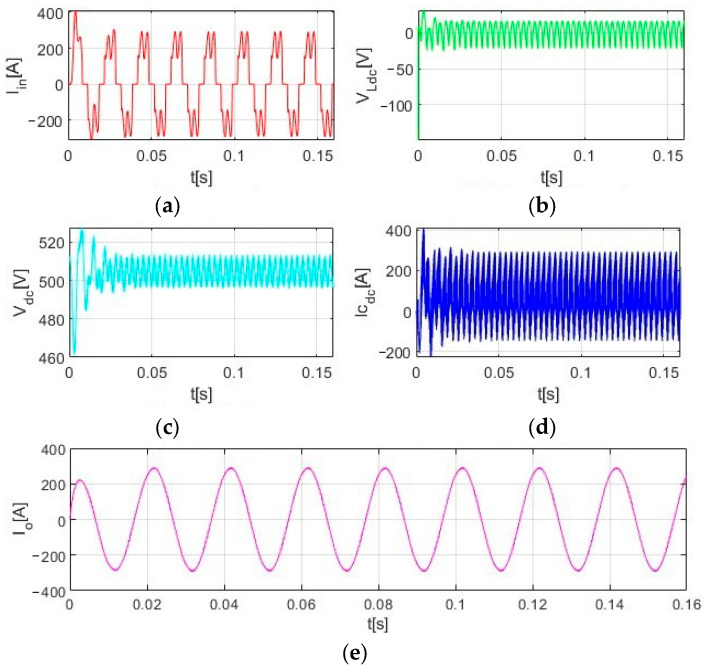
Output current waveform. (**a**) Input current; (**b**) Voltage of DC reactor; (**c**) Bus voltage; (**d**) Ripple current of bus capacitor; (**e**) Output current.

**Figure 9 sensors-23-08211-f009:**
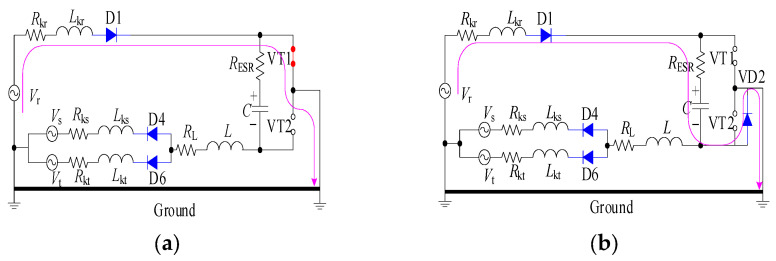
Energy transmission of VT1 turns on and turns off. (**a**) The equivalent circuit of VT1 turns on; (**b**) The equivalent circuit of VT1 turns off.

**Figure 10 sensors-23-08211-f010:**
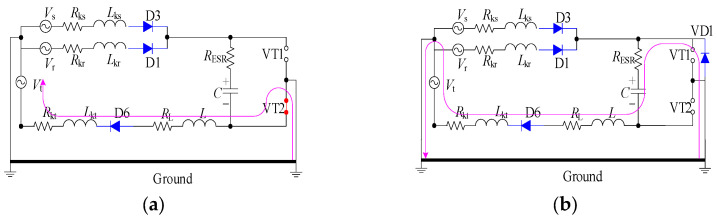
Energy transmission of VT2 turns on and turns off. (**a**) The equivalent circuit of VT2 turns on; (**b**) The equivalent circuit of VT2 turns off.

**Figure 11 sensors-23-08211-f011:**
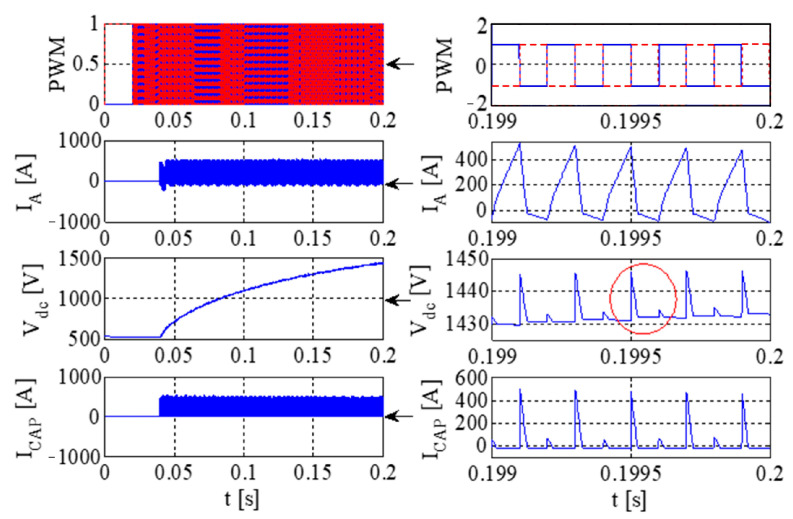
The PWM waveform, output current, DC bus voltage waveform, and capacitor current waveform of IGBT.

**Figure 12 sensors-23-08211-f012:**
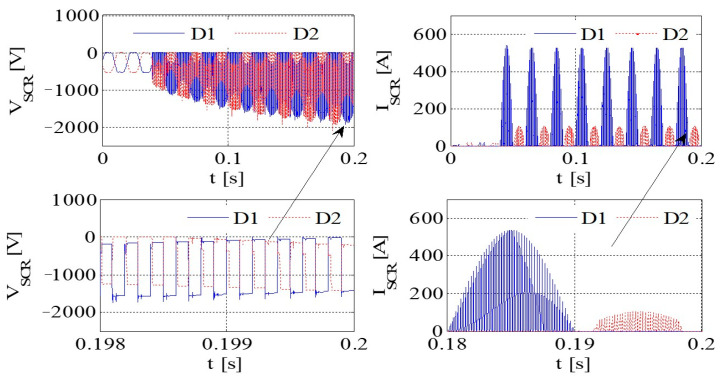
The output voltage and output current of a bridge arm of the rectifier.

**Figure 13 sensors-23-08211-f013:**
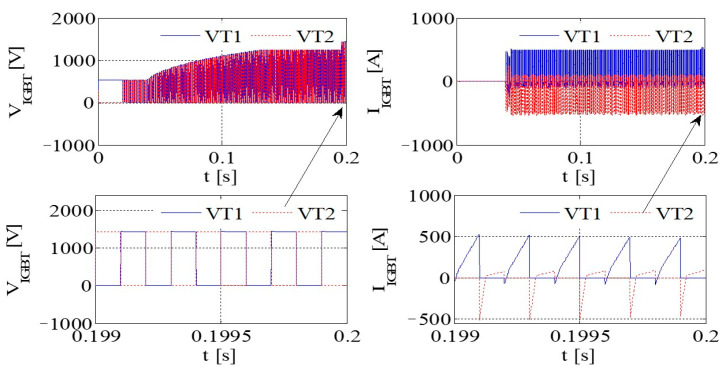
The output voltage and current of an IGBT bridge arm on IGBT.

**Figure 14 sensors-23-08211-f014:**
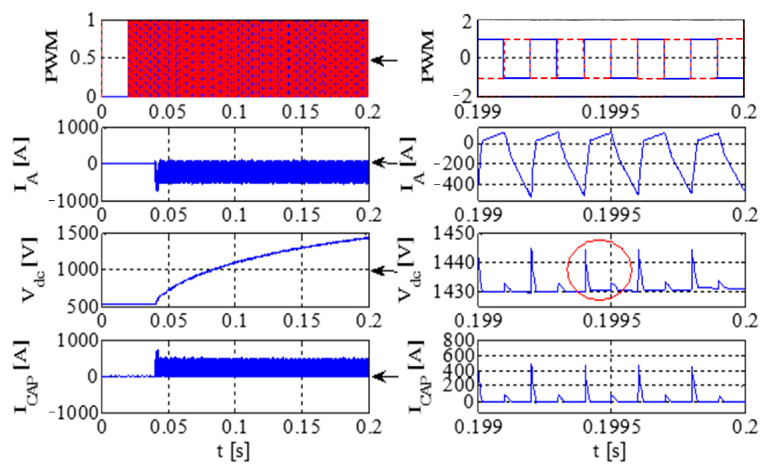
IGBT PWM waveform, output current, DC bus voltage waveform, and capacitor current waveform.

**Figure 15 sensors-23-08211-f015:**
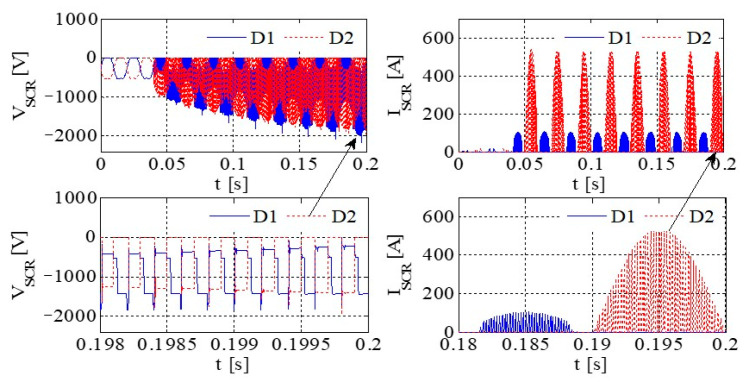
The voltage and current at the output of a rectifier bridge arm.

**Figure 16 sensors-23-08211-f016:**
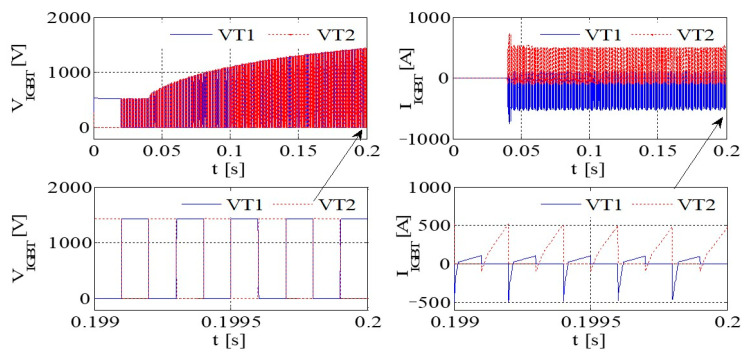
The voltage and current at the output of an IGBT bridge arm.

**Figure 17 sensors-23-08211-f017:**
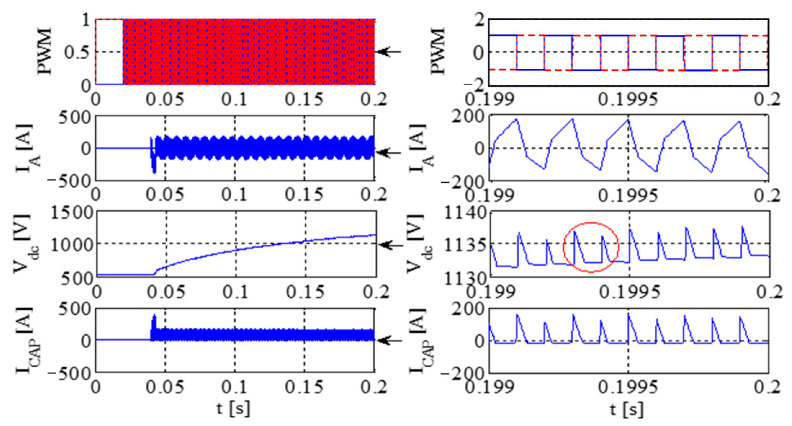
The PWM waveform of an IGBT, the output current, the DC bus voltage waveform, and the capacitor current waveform.

**Figure 18 sensors-23-08211-f018:**
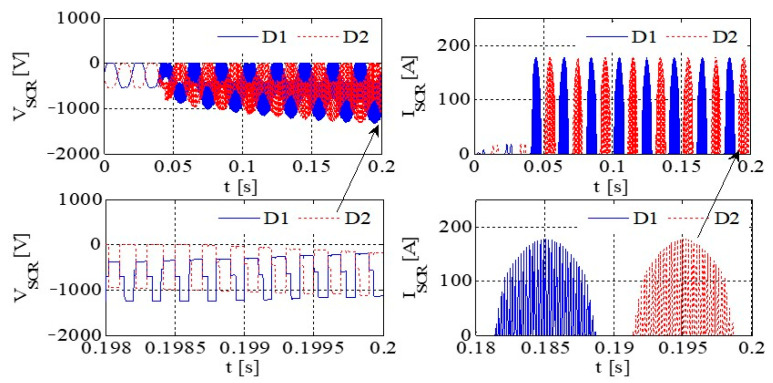
The output voltage and current of a bridge arm on rectifier.

**Figure 19 sensors-23-08211-f019:**
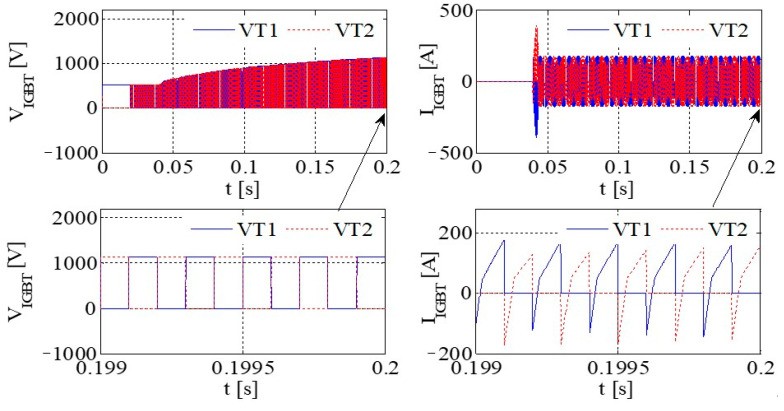
The output voltage and current of a bridge arm on IGBT.

**Figure 20 sensors-23-08211-f020:**
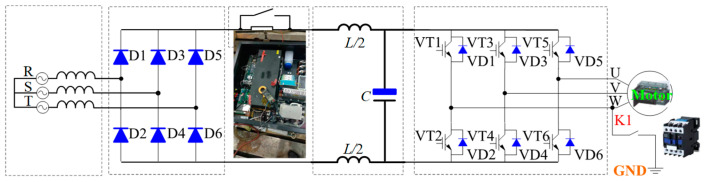
Experimental circuit diagram.

**Figure 21 sensors-23-08211-f021:**
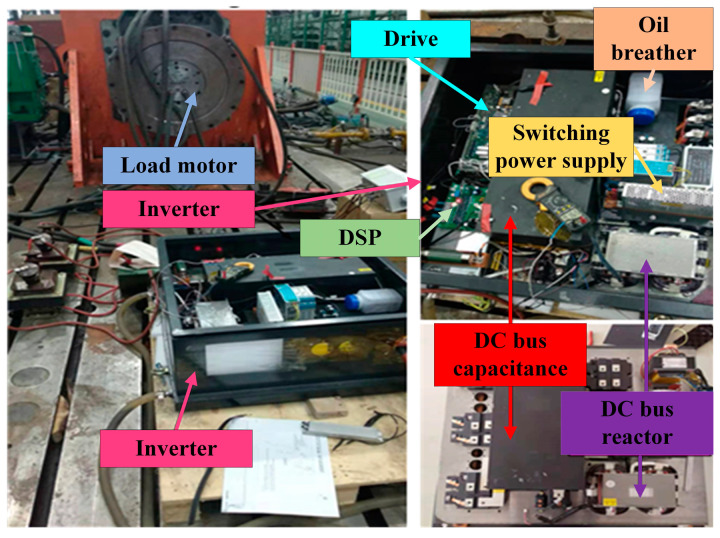
Inverter, permanent magnet motor, and experimental system.

**Figure 22 sensors-23-08211-f022:**
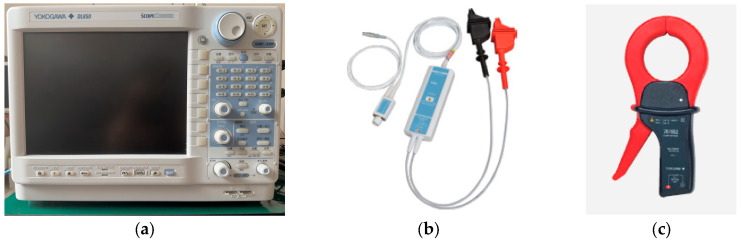
Experimental measurement equipment: (**a**) YOKOGAWA/DL850 oscilloscope; (**b**) High voltage differential probe; (**c**) Current sensor.

**Figure 23 sensors-23-08211-f023:**
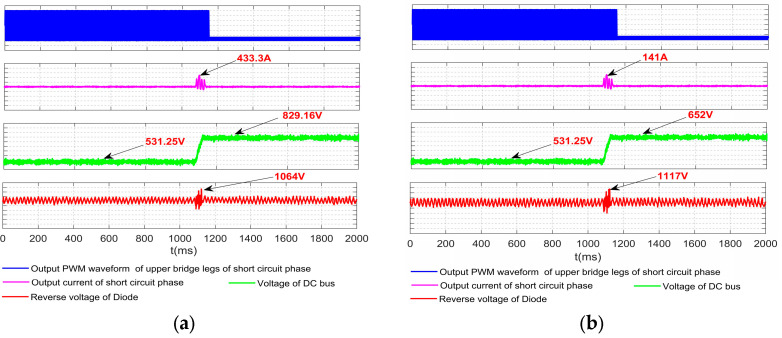
Experimental results waveform (**a**) DC reactor on the cathode of the DC bus; (**b**) DC reactor on the anode and cathode of the DC bus.

**Table 1 sensors-23-08211-t001:** Unit of measurement and corresponding abbreviations.

Full Name	Abbreviation	Full Name	Abbreviation
Volt	V	Ampere	A
Ohm	Ω	Watt	W
Hertz	Hz	hour	h
Revolutions Per Minute	rpm	Celsius centigrade	°C
Meter	m	Tesla	T

**Table 2 sensors-23-08211-t002:** 380 V/110 kW converter system parameters.

Project	Value	Project	Value
Rated voltage	380–500 V	Maximum limited current multiple	2.5
Rated current	205 A	Overload multiple	1.1
Rated power	110 kW	Overload time	1 min
Rated frequency	50 Hz	Coolant	Air
Switch frequency	3.6 kHz	Ambient temperature	40 °C
Rated speed	1478 rpm	Altitude	0~1 km

**Table 3 sensors-23-08211-t003:** Calculation results of bus capacitance loss.

Project	Single Bus Capacitance	Total Bus Capacitance	Bus Capacitor Life
Rated load	18.4268 W	110.5610 W	27,418 h
Overload	18.4209 W	110.5253 W	13,698 h
Under load	18.4278 W	110.5670 W	—

**Table 4 sensors-23-08211-t004:** The upper limit of the inductance value of the DC reactor.

Project	The Inductance Value of the DC Reactor
*K*_ovd_ = 1	306.592 μH
*K*_ovd_ = 2	369.714 μH

**Table 5 sensors-23-08211-t005:** The lower limit of inductance value of the DC reactor.

Project	The Inductance Value of the DC Reactor
*K*_ovd_ = 1	227.2586 μH
*K*_ovd_ = 2	263.1879 μH

**Table 6 sensors-23-08211-t006:** Size parameters of reactor core.

Project	Value	Project	Value
A	133.2 mm	F	66.6 mm
B	111 mm	G	88.8 mm
C	44.4 mm	H	22.2 mm
D	22.2 mm	Thick	0.35 mm
E	22.2 mm		

**Table 7 sensors-23-08211-t007:** Equivalent series resistance and loss of each part of the DC reactor.

Application	Rated Load	Overload	Under Load
General application	119.2302	142.6903	116.6327
Heavy-duty application	83.0329	177.0670	72.7752
*ESR_Ldc_*	1.9 mΩ		

**Table 8 sensors-23-08211-t008:** Simulation results of different inverter topologies.

Item	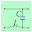	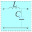	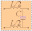
I_A_ [A]	542	−537.5	180
V_DC_ [V]	1443	1443	1135
I_CAP_ [A]	512	497	165
V_D1_and V_D2_ [V]	1890	1920	−1890	−1920	1320	1337
I_D1_ and I_D2_ [A]	537	104	537	104	177	177
VT1 and VT2 [V]	1446	1435	1435	1435	1135	1135
I_VT1_ and I_VT2_ [A]	542	−542	−542	542	180	180

**Table 9 sensors-23-08211-t009:** The effective average current and rated effective current of the rectifier.

Application	Rated Average Current	Rated RMS Current
General application	119.2302	142.6903
Heavy-duty application	83.0329	177.0670

**Table 10 sensors-23-08211-t010:** The measurement conditions of this experiment.

Item	Parameter
Experimental ambient temperature	25 °C
Experimental ambient humidity	45% RH
Frequency bandwidth	150 MHz
Sampling rate	500 MHz
Voltage probe measurement range	−5000~5000 V (DC + ACpeak)
Measurement accuracy of voltage probe	Error in the range of ±400 V: ±2%; error in the range of ±1000 V: ±3%
Measurement range of current probe	0~1000 A
Measurement accuracy of the current probe	DC ± (0.05% of rdg + 30 μA) 50/60 Hz ± (0.05% of rdg + 30 μA)

**Table 11 sensors-23-08211-t011:** The experimental results of inverter one-to-ground short circuit.

DC Reactor Location	Output Current of Short Circuit Phase	Voltage of DC Bus	Reverse Voltage of Diode
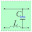	433.3 A	829.16 V	1604 V
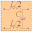	141 A	652 V	1117 V

## Data Availability

Not applicable.

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
