# Peer review of "Study of the Protection and Energy Transmission Modes of One Phase Short Circuit to Ground in Inverters"

_sensors, 2023, doi:10.3390/s23198211_

Round 1

Reviewer 1 Report

The idea of the paper is quite good enough but with some modifications and suggestions can further go to next level.

1.      The paper is too long. Should be shortened.

2.      Many typo-errors are there, as well many grammatical and English errors are there. Recheck the text completely.

3.      The abstract should be corrected in terms of the paper novelties, adding some numerical results, and used software.

4.      Introduction section needs to be rewritten. Bulk-referencing should be avoided. Authors are advised to cite some recent references in the article.

5.      Literature review should be enriched by adding some detailed comparisons regarding the pervious method and what is discussed in this work.

6.      Figure 2. Calculation process of bus capacitance parameter: Call the Newton Raphson program, For what purpose. Not clear.

7.      Many of formulas are clear, and there is need to mentioning all of them. Delete them, and instead refer to relevant works.

8.      Table 2. Calculation results of bus capacitance loses. Bus capacitor life. The calculation method is not clear.

9.      Equations (30), and (31) are not clear. Should be re- rewritten.

10. Conclusion needs to be rewritten

11. The simulation results section needs to be rechecked. Many typo-errors are there in that particular section.

The idea of the paper is quite good enough but with some modifications and suggestions can further go to next level.

1.      The paper is too long. Should be shortened.

2.      Many typo-errors are there, as well many grammatical and English errors are there. Recheck the text completely.

3.      The abstract should be corrected in terms of the paper novelties, adding some numerical results, and used software.

4.      Introduction section needs to be rewritten. Bulk-referencing should be avoided. Authors are advised to cite some recent references in the article.

5.      Literature review should be enriched by adding some detailed comparisons regarding the pervious method and what is discussed in this work.

6.      Figure 2. Calculation process of bus capacitance parameter: Call the Newton Raphson program, For what purpose. Not clear.

7.      Many of formulas are clear, and there is need to mentioning all of them. Delete them, and instead refer to relevant works.

8.      Table 2. Calculation results of bus capacitance loses. Bus capacitor life. The calculation method is not clear.

9.      Equations (30), and (31) are not clear. Should be re- rewritten.

10. Conclusion needs to be rewritten

11. The simulation results section needs to be rechecked. Many typo-errors are there in that particular section.

Author Response

Thank you very much for your proposed amendments. I have been modified in accordance with the opinions, please see the annex for details.

Reviewer 2 Report

The paper „Study of the Protection and Energy Transmission Modes of One Phase Short Circuit to Ground in Inverters” reports a study regarding a phase-to-ground short circuit situation of LC filter frequency transformer. The authors propose to solving this problem first of all by using an analytical method for calculating the capacity of the DC bus of the inverter as well as the parameters of the DC reactor. The analytical study is clear and solid. Then, the authors propose a mathematical model of the double boost circuit when the inverter is single-phase short-circuited, taking into account the influence of the leakage inductance of the power transformer. By considering the short circuit to ground in one phase of inverters is proposed a novel method for diodes on rectifier section and IGBT transistors on the inverter section determination. Effectiveness of the proposed method is verified and by experiment.

Following the review of the paper "Study of the Protection and Energy Transmission Modes of One Phase Short Circuit to Ground in Inverters", we can conclude that:

1.      A correctly and also complete formulated nomenclature, which containing all the physical quantities involved in the paper along with the measurement units in the international system, is necessary to be introduced in the paper. Thus, the readers can more easily clarify physical quantities and abbreviations involved in the paper. In adittion, the abbreviations should be explained when they first appear in this paper;

 2.      It is necessary for the authors to improve in the paper the discussion regarding the PWM waveforms used to control the inverter section and specify their frequency considered for simulation;

 3.       It is necessary to present in the paper the whole scheme of measurements and experimentation used to obtain all the experimental values that are presented  in the paper. In this regard, it can be said that Figure 20  it is not relevant. Additional figures in this respect would be desirable, if possible with the real representations of the measuring devices;

 4.      Please make Figure 20 more readable;

 5.      It is necessary to present in the paper what were the criteria for choosing the frequency converter 380V/110kW type. Please explain a little bit more this. Also, recent regulations recommend 400V/110kW;

 6.      Regarding the experimental results, it is necessary for the authors to indicate in the paper, all type of the equipment that is used in order to obtain the experimental values. Also, it is necessary for the authors to improve the discussion regarding the calibration of the measurement system;

 7.      It is necessary for the authors to improve in the paper the discussion regarding the reproducibility of the obtained values through measurements. Also, it is necessary to clarify in the paper all the measurement conditions for the experiments performed;

 8.      In general, it is elegant that a paragraph does not not to start with a figure but with a comment. In this regard, is necessary to add a comment before the Figure 20. Also, it is elegant that a paragraph does not end with a figure or table, but with a comment (Figure 6, Table 2, etc);

 9.      A parallel made between the results obtained experimentally and those obtained by numerical simulation would be desirable. It is necessary for the authors to clarify this important aspect;

10.  It is necessary that the paper be completed by a chapter on Conclusions. The conclusions of the paper must be comprehensive and well-organized information; the paper contains many results that need to be highlighted in the Conclusions chapter. The conclusions of the paper must contain the possible implications of these study in future practical developments. What are the prospects for capitalizing on this research? It is necessary to add and these aspects to the conclusions.

Minor editing of English language required.

Author Response

(The authors gave the same response as above.)

Round 2

Reviewer 1 Report

The authors have done good work. There is no further comment.

Minor editing of English language required

Author Response

I appreciate your review. In the resubmitted document, I made further modifications to the English language. Thank you very much for your suggestion.

Reviewer 2 Report

The paper „Study of the Protection and Energy Transmission Modes of One Phase Short Circuit to Ground in Inverters” reports a study regarding a phase-to-ground short circuit situation of LC filter frequency transformer. The authors propose to solving this problem first of all by using an analytical method for calculating the capacity of the DC bus of the inverter as well as the parameters of the DC reactor. The analytical study is clear and solid. Then, the authors propose a mathematical model of the double boost circuit when the inverter is single-phase short-circuited, taking into account the influence of the leakage inductance of the power transformer. By considering the short circuit to ground in one phase of inverters is proposed a novel method for diodes on rectifier section and IGBT transistors on the inverter section determination. Effectiveness of the proposed method is verified and by experiment.

The paper, in its revised form is not significantly improved, the authors basically answered several points in a superficial way.

Following the review of the paper "Study of the Protection and Energy Transmission Modes of One Phase Short Circuit to Ground in Inverters", in the revised form, we can conclude that:

1.      The nomenclature is missing in the revised form of the paper. A correctly and also complete formulated nomenclature, which containing all the physical quantities involved in the paper along with the measurement units in the international system, is necessary to be introduced in the paper. Thus, the readers can more easily clarify physical quantities and abbreviations involved in the paper. In adittion, the abbreviations should be explained when they first appear in this paper;

 2.      The answer to this point cannot be found in the revised form of the paper. It is necessary for the authors to improve in the paper the discussion regarding the PWM waveforms used to control the inverter section and specify their frequency considered for simulation;

 3.      It is necessary to present in the paper the whole scheme of measurements and experimentation used to obtain all the experimental values that are presented  in the paper. In this regard, it can be said that Figure 20 it is not relevant. Additional figures in this respect would be desirable, if possible with the real representations of the measuring devices. In addition, Figures 20 and 21 have the same title;

 4.      Regarding the experimental results, it is necessary for the authors to indicate in the paper, all type of the equipment that is used in order to obtain the experimental values. Also, it is necessary for the authors to improve the discussion regarding the calibration of the measurement system;

 5.      It is necessary for the authors to improve in the paper the discussion regarding the reproducibility of the obtained values through measurements. Also, it is necessary to clarify in the paper all the measurement conditions for the experiments performed;

 6.      In general, it is elegant that a paragraph does not end with a figure or table, but with a comment. In this regard, is necessary to add a comment after the Figure 6, and Table 2.

 7.      A parallel made between the results obtained experimentally and those obtained by numerical simulation would be desirable. The parallel must be made by indicating the actual values obtained and not by a conclusion-type phrase. It is necessary for the authors to clarify this important aspect;

 8.      The Conclusions chapter introduced in the revised form of the paper is extremely short. The conclusions of the paper must be comprehensive and well-organized information; the paper contains many results that need to be highlighted in the Conclusions chapter. The conclusions of the paper must contain the possible implications of these study in future practical developments. What are the prospects for capitalizing on this research? It is necessary to add and these aspects to the conclusions.

Minor editing of English language required.

Author Response

I appreciate your proposed amendments, the paper has been modified in accordance with your recommendations, reply details please see the annex. Thank you very much again.

Round 3

Reviewer 2 Report

Following the review of the paper Study of the Protection and Energy Transmission Modes of One Phase Short Circuit to Ground in Inverters, in the present third revised form, I can conclude that:

  I have looked carefully on the answers of the authors, regarding on all recommendations. In addition, I have looked carefully of the entire paper, in the present third revised form. I think that the authors solved correctly the majority of the recommendations. Also, have been introduced within this third revised form of the paper the result paragraphs.

 However, some recommendations are necessary.

 1.      In general, it is elegant that a paragraph does not end with a figure, but with a comment. In this regard, is necessary to add a comment after Figure 1, Figure 8 and Figure 16;

2.      Regarding the Figure 20, it would be desirable to reconsider the title of the figure;

3.      The captures made by the digital oscilloscope and illustrated in Figure 21 should be presented much more legibly (to be presented in such a way that all signals from all channels can be distinguished very well), illustrated with Figure 21 (c) and Figure 21 (d) respectively and commented in the text of the paper;

4.      It is necessary to clarify in the paper all the measurement conditions for the experiments performed;

Author Response

Thank you very much for the suggestions made by the reviewers. The article has been modified according to your suggestions. Please refer to the attachments and papers for details.
